# Distribution of Deer Keds (Diptera: Hippoboscidae) in Free-Living Cervids of the Tuscan-Emilian Apennines, Central Italy, and Establishment of the Allochthonous Ectoparasite *Lipoptena fortisetosa*

**DOI:** 10.3390/ani11102794

**Published:** 2021-09-25

**Authors:** Annalisa Andreani, Laura Stancampiano, Antonio Belcari, Patrizia Sacchetti, Riccardo Bozzi, Maria Paola Ponzetta

**Affiliations:** 1Department of Agriculture, Food, Environment and Forestry (DAGRI), University of Florence, Piazzale delle Cascine, 18-50144 Firenze, Italy; antonio.belcari@unifi.it (A.B.); patrizia.sacchetti@unifi.it (P.S.); riccardo.bozzi@unifi.it (R.B.); mariapaola.ponzetta@unifi.it (M.P.P.); 2Department of Veterinary Medical Sciences, University of Bologna, Via Tolara di Sopra, Ozzano dell’Emilia, 50-40064 Bologna, Italy; laura.stancampiano@unibo.it

**Keywords:** deer, insects, parasitism, invasive species, *Lipoptena cervi*

## Abstract

**Simple Summary:**

In recent years, the increased presence of wildlife in habitats close to urban settlements has raised concerns about the risk of pathogen transmission from wild animals to humans due to the spread of different parasites. For this reason, a survey aimed at describing the dispersal and parasitism level of two cervid ectoparasites was carried out in the northern Apennines, in central Italy. The presence of two hippoboscids, the autochthonous *Lipoptena cervi* and allochthonous *L. fortisetosa*, native to Eastern Asia and recently recorded in Italy, were assessed on their main host species (red deer, fallow deer, and roe deer), considering host sex and age. The alien species *L. fortisetosa* was found to be widespread in the study area, most likely competing with *L. cervi*. Moreover, red deer seemed to be the favored host of both flies, with differences in sex and age class preferences. This study demonstrated the importance of regularly monitoring the populations of these parasites, especially the invasive species, due to the risks to human health, as these insects are potential vectors of pathogens.

**Abstract:**

*Lipoptena fortisetosa* and *L. cervi* are hematophagous ectoparasites belonging to the Hippoboscidae family and preferentially living on cervids. In recent years, they have received specific attention due to the great increase in the abundance of their host species, and to their medical and veterinary importance as possible vectors of pathogens harmful to humans and animals. The aim of this study was to investigate the parasitism level of both of these flies on their main hosts in Italy, which are red deer, fallow deer, and roe deer, and to highlight a possible preference for a species, sex, or age class among the hosts. Deer keds were collected by examining 326 cervids hunted in the Tuscan-Emilian Apennines. Outcomes showed that *L. fortisetosa* has greatly spread throughout the study area, where it competes with the autochthonous *L. cervi*. Moreover, red deer was the favored host species of both ectoparasites, while different preferences for host sex and age classes were observed in the two hippoboscids. The regular monitoring of deer ked populations, especially the allochthonous *L. fortisetosa*, which is continuously spreading in Europe, is recommended to expand the knowledge on these parasitic species that are potentially dangerous to public health.

## 1. Introduction

In recent years, a substantial expansion range of free-living ungulate species has occurred in many European countries [1], including Italy [2,3] and particularly the Tuscany region in the northern Apennines [4], leading to a consequent increase in the abundance of their ectoparasites, which can potentially colonize and adapt to new territories and host species.

Particularly notable is the spread of allochthonous ungulates and their ectoparasites in new countries, which demonstrates the adaptability of some alien species with the consequent risk of competition with native animals and a compromised ecosystem balance. In this respect, both the hippoboscid *Lipoptena fortisetosa* Maa, 1965 and its original host *Cervus nippon* (sika deer) have been recently reported in Italy [5,6].

Members of the genus *Lipoptena* (Diptera: Hippoboscidae) are obligate hematophagous ectoparasites that permanently live on a restricted range of hosts, especially Cervidae [7,8]. These flies attack several species, referred to as “accidental hosts” or “feeding hosts”, that are used as food sources only, but they are able to successfully thrive only on a few mammals, referred to as “definitive hosts” or “breeding hosts”, which have the requisites to guarantee the reproduction and survival of these flies [9,10].

Parasites establish a close association with their suitable hosts through morphological and physiological adaptations [11,12]. Females are viviparous and give birth to fully grown larvae, one at a time, that thereafter pupate and fall from the host to the ground due to the deers’ movements. Reproduction occurs all-year-round, but the emergence of newly winged adults takes place from late spring to autumn. Flies spend the first period as imago searching for a suitable host to settle on for their whole life. Subsequently, they crawl into the fur of the animal and gradually shed wings through a horizontal predetermined breaking line as a result of their passage between the hairs of the host [10]. When the ectoparasites become wingless, they are no longer able to fly, making it quite difficult to switch to other subjects; thus, they strictly depend on the selected host. Nevertheless, moving to other specimens is possible, especially from cervid females to their fawns and vice versa, or moving can occur during allogrooming behavior among deer [13].

In Italy, the species of the Lipopteninae subfamily infesting deer are *Lipoptena cervi* (Linnaeus, 1758) and *L. fortisetosa* (named deer keds). In this country, this adventive ectoparasite has also been collected from other cervid species, demonstrating its ability to successfully colonize different hosts [6]. *Lipoptena cervi* is widely distributed in more than 20 European countries [14] and has spread across North America since the beginning of the twentieth century [15]. *Lipoptena fortisetosa* is native to Japan but has spread, as far as we know, to at least 12 European countries [16], including Italy. Both of these species show a preference for parasitizing Cervidae: *L. cervi* has been recorded on *Cervus elaphus*, *Dama dama*, *Alces alces*, *Capreolus capreolus*, and *Moschus moschiferus*, while *L. fortisetosa*, although it was considered quite restricted to its original host, *Cervus nippon*, has also been collected from *Cervus elaphus*, *Capreolus pygargus* and *Capreolus capreolus* [15,17,18]. *Lipoptena cervi* and *L. fortisetosa* can concurrently be found on the same deer [6,19], together with ticks. Hippoboscids can heavily infest hosts, compromising them physically and behaviorally [20,21]. Hosts are annoyed by keds, especially because of their movements in the hosts’ fur and their recurrent feeding on blood, up to 20 times per day for each fly of both sexes. Severe attacks are mainly documented on moose, on which tens of thousands of *L. cervi* have been collected from a single individual host [20,22]. Such infestations directly harm the hosts, causing issues such as skin inflammation, injuries, and blood loss leading to possible secondary infections. Observing the discoloration caused by fresh blood loss in moose bedding sites, Kaunisto et al. [23] showed that a high number of deer keds can cause bleeding in their hosts, leading to capillary vein and skin damage. In the case of extreme harassment, physiological and behavioral changes can be observed in reindeer, as reported by Kynkäänniemi et al. [21]. These authors verified that heavy parasitism induces hosts to react with defense actions, such as shaking their head and body or stamping feet, reducing the time spent grazing and causing a decrease in body weight and welfare.

Usually, hippoboscids infest animal hosts, but they can also bite humans, creating a consequent health risk, which needs to be verified with further studies, that these insects may transmit some zoonotic pathogens [24,25,26,27,28,29,30]; however, no overt form of these diseases has yet been detected in deer hosts. Moreover, the bites of deer keds on humans can result in persisting and itching papules, in addition to dermatitis [31,32,33,34]. In countries where ked density is particularly high, people frequenting forests and natural areas complain of the great nuisance of keds, which ultimately leads to a reduction in recreational activities and hunting in this habitat [32].

Understanding the relationship between animals and their parasites is crucial; in fact, the spread of deer keds seems to be strongly related to the availability and density of potential hosts, whose spatiotemporal variation is considered one of the most important factors affecting the dispersal of ectoparasites [35]. Other factors also affect the presence and distribution of these insects, such as cold tolerance, habitat, and predation [22]. In particular, climate change is suggested to support the increase in deer ked populations, since temperature has a positive effect on the duration of the host-seeking period, extending the possibility of host acquisition [36,37]. The risks related to the general increase in the density of cervid populations and global warming, which could facilitate the expansion of ectoparasites, make further investigations on these flies necessary.

*Lipoptena cervi* and *L. fortisetosa* have been recently investigated under different points of view, such as morphology, distribution, or disease transmission, but no studies on the parasitic dynamics of these flies related to their hosts have been carried out in Italy to date.

The aim of this research was to provide an insight into the presence and epidemiology of *L. fortisetosa* on cervids (*Cervus elaphus*, *Dama dama*, and *Capreolus capreolus*) in some areas of the Tuscan-Emilian Apennines, central Italy, and any differences in infestation with respect to the native parasite, *L. cervi*.

In particular, the goals were to investigate (a) the distribution of these parasitic flies in the study area; (b) the level of infestation on the three examined ungulate species; (c) the possible preference of the ectoparasites for a definite host species; and (d) the potential predilection of deer keds toward host sex and age class.

## 2. Materials and Methods

### 2.1. Study Area

Samples were collected in the Tuscan-Emilian Apennines (central Italy) from an area extending along approximately 207 km of surrounding territories encompassing four provinces of Tuscany (Arezzo, Florence, Pistoia, and Prato) and three provinces of Emilia Romagna (Bologna, Modena, and Reggio Emilia). In the study area, the landscape is characterized by different altitudes, with hills and mountains ranging from 500 to a maximum of approximately 2000 m. This large region is covered by a variety of vegetation zones, and inhabited areas of different extents intersect valleys. In general, large cervid populations, managed through wildlife hunting programs, exist in the study area.

### 2.2. Sample Collection and Taxonomic Identification of Hippoboscidae

Deer keds were collected from the fur of free-ranging cervids of three different species: *Cervus elaphus*, *Capreolus capreolus*, and *Dama dama* (Figure 1). All ectoparasites were obtained from animals hunted during the culling seasons of 2018–2020. The examined cervids were sampled almost continuously, depending on the specific hunting period, from November 2018 to March 2020, except for May 2019. A total of 326 animals (181 red deer, 107 roe deer, and 38 fallow deer) were sampled, and insects were picked off from two pieces of cervid skin that were voluntarily provided by hunters and local wildlife technicians, who were carefully instructed of this process before the hunting period. They were required to cut two skin samples, 20 × 20 cm each, one from the neck and the other from the groin region [10,38], and to store them in a plastic bag at −20 °C until they were transferred to the staff of the Department of Agriculture, Food, Environment and Forestry (DAGRI), University of Florence.

Each skin sample was accompanied by a form containing detailed information such as animal species, sex, and age class. Age classes were the following:Fawn (<1 year for all cervid species);Subadult (between 1 and 4 years for fallow deer; 1 and 2 years for roe deer; 1 and 3 years for female red deer and 1 and 5 years for male red deer);Adults (>4 years for fallow deer; >2 years for roe deer; >3 years for female red deer and >5 years for male red deer).

Fur samples were thawed and then visually examined for deer keds. All the parasites were manually removed with forceps and morphologically observed under a stereomicroscope (Leica/Wild MZ16, equipped with an L2 illuminator; Leica Microsystems, Wetzlar, Germany) for taxonomic identification using keys and previously described characters [6,9,39,40].

Subsequently, all the insects were separated by species and sex, counted, and stored in 70% ethanol or frozen at −20 °C, pending further analyses.

All cervid handling procedures followed the regional, national, and institutional guidelines.

Data were transcribed and reported into different software programs: QGIS 3.16.3 Hannover (QGIS Geographic Information System, open source software available at http://www.qgis.org, accessed on 25 Septmber 2021); GRASS 7.8.5 (Geographic Resources Analysis Support System, open source software available at grass.osgeo.org, accessed on 25 Septmber 2021), and Excel (Microsoft 365, 2016, Microsoft Italia, Milano, Italy) to be used for further analyses.

From here onward, any mention of parasites per host animal is always referred to as the sum of the keds collected from the two skin samples described above.

### 2.3. Parasitological Index

The infestation of deer keds on the three different host species was described using parasitological indices according to Margolis et al. [41]. All the descriptors were stratified by host and ectoparasite species. The distribution of *Lipoptena* spp. was evaluated by the parasitological index of density (average number of parasites per unit area (cm^2^) of the host body), prevalence (percentage of infested deer); abundance (average number of parasites per host), mean intensity (average number of parasites per infested host), and minimum and maximum intensity (*I*min-*I*max). Moreover, the variance-to-mean ratio (variance of infestation divided by mean abundance) was calculated as the aggregation index, considering the distribution as overdispersed (or aggregated) if the value was >1, as demonstrated by Barbour and Pugliese [42].

### 2.4. Statistical Analyses

Data were statistically analyzed using R software [43]. Preliminary analysis highlighted aggregated parasite distribution; therefore, generalized linear models (in particular negative binomial regression) with the abundance of each parasite species as the dependent variable were built using the MASS package [44,45,46].

First, the chi-square test was used to compare parasite prevalence among the three host species.

Differences in parasite abundance among the three host species were evaluated using a univariable model to determine the primary host species; therefore, multivariable models were used to evaluate the influence of host-related variables on parasite abundance in the formerly determined host species.

## 3. Results

Out of the 326 examined cervids, 287 harbored deer keds (88.0%). The morphological analyses of the 23,074 collected flies revealed the presence of two hippoboscid species, identified as *L. fortisetosa* and *L. cervi*. Of the total insects, 18,441 were *L. fortisetosa**,* and 4633 were *L. cervi*; even though the total highest number of insects was *L. fortisetosa*, some host animals were more infested with *L. cervi*. Of the total examined hosts, 127 cervids carried both the *Lipoptena* species, while 26 out of 107 tested roe deer, 45 out of 181 tested red deer, and two out of 38 tested fallow deer did not harbor any parasites. The data on the overall deer ked infestations in the three host species are given in Table 1.

The number of males and females of the two ectoparasites, together with the values of the parasitological parameters for each of the three host species, are reported in Table 2.

The highest number of deer keds on a host subject was found in red deer, while the number of parasites obtained from roe deer and fallow deer was much lower. A maximum of 1,844 *L. fortisetosa* were collected from a host specimen, while a maximum of 398 flies of *L. cervi* were picked off a single red deer.

The chi-square test highlighted significant differences among the prevalence of both parasite species in the three hosts (*p* = 0.000 and *p* = 0.000, respectively for *L. fortisetosa* and *L. cervi*). In particular, fallow deer showed the highest prevalence for *L. fortisetosa* (94.8%), while it displayed the lowest prevalence value for *L. cervi* (21.5%). Although fallow deer was infested more often with *L. fortisetosa* than the other two hosts, the abundance of this parasite was highest for red deer, as confirmed by the univariable negative binomial regression (*p* = 0.000). Additionally, *L. cervi* abundance was higher in red deer than in the other two host species (*p* = 0.000). Notably, only 17 *L. cervi* were found on the 38 analyzed fallow deer (Table 2).

The aggregation index was >1 for all the cervid species for both keds, meaning that the parasites were aggregated over the host populations, as illustrated in Figure 2.

Further details on the epidemiological parameters stratified by the sex and age of the hosts are provided in Table 3.

The multivariable negative binomial regression, taking into consideration sex and age, was constructed for red deer only since it was the primary host species for both parasites. The results are reported in Table 4 and Table 5 for *L. cervi* and *L. fortisetosa*, respectively. *Lipoptena cervi* was significantly less abundant in females than in males and in fawns than in subadults, while *L. fortisetosa* was significantly less abundant in adults than in subadults. The interactions between sex and age classes were significant for both parasite species, as evident in Figure 3. The plots of the residuals in Figure 4 show a good residual pattern, with similar residual distributions across the levels of the predicted values.

## 4. Discussion

This study documents the presence of both native *L. cervi* and allochthonous *L. fortisetosa* in the Tuscan-Emilian Apennines (central Italy). These hippoboscids have already been documented in Italy, but the literature is still limited to local areas [6]. *Cervus elaphus*, *C. capreolus,* and *D. dama* were hunted and sampled for ectoparasites, revealing a considerable distribution of these flies. All three host species were infested with both parasites, showing the adaptability of these parasites to the examined cervids. Although *L. cervi* seems to have a greater worldwide distribution than that of *L. fortisetosa* [47], our survey proves that locally allochthonous species may be were largely more abundant than autochthonous species, demonstrating that the introduced *L. fortisetosa* is numerous in the study area and strongly competes with native hippoboscids which not only live in the same geographic territories but also share the same host species. Our study confirms the coexistence of *L. cervi* and *L. fortisetosa* in the same area, as evidenced in other European regions, such as northeastern Poland and Lithuania [19,48]. Moreover, *L. fortisetosa* was found to share the same host with other dipteran ectoparasite species in Japan, where it was sampled on Japanese deer with *Lipoptena sikae* [49].

Although many cervid species have been reported as suitable definitive hosts for *L. cervi*, red deer and moose seem to be the favored species in Europe, while the Japanese deer, *C. nippon,* is considered the main and original host for *L. fortisetosa* [15]. On red deer, *L. cervi* can reach a very high frequency of infestation, ranging between 78% and 100% [19,50], while it is less abundant on fallow deer [10]. A heavy infestation of *L. cervi* in four hunted roe deer was recorded in Romania, with the average number of flies exceeding 2500 parasites per host [51]. Nevertheless, in other countries, a lower infestation prevalence was noted for *L. cervi* on the same host species, varying from 36% to 64% [52,53,54]. In Lithuania, *L. cervi* was less abundant on roe deer specimens than on in the other two examined host species, red deer and moose [19]. Additionally, *L. fortisetosa* showed a preference for attacking red deer over roe deer since the prevalence of infestation was 49% [55] or 100% [19], and 23% [54] or 90% [19], respectively. To the best of our knowledge, no studies balancing *L. fortisetosa* infestation on red deer, roe deer, and fallow deer have been carried out, but our data are consistent in stating that both parasites prefer red deer hosts over the other two cervids.

Interestingly, although some host species showed a high overall fly infestation prevalence (i.e., 94.7% *L. fortisetosa* on *D. dama*), no one species reached the 100% prevalence recorded on moose by several authors [19,38,56]. Moreover, in our survey, the overall density of *Lipoptena* spp. was higher on red deer than on the other two host species, at 0.12/cm^2^ for *L. fortisetosa* and 0.03/cm^2^ for *L. cervi*. Yet, a much greater number of deer keds was counted on moose, on which these parasites reached as many as 17,500 specimens on a single bull [56]. Since moose generally harbors a large number of hippoboscids, we could deduce that this species is more suitable for these parasites. Kadulski [52] found that among moose, red deer, roe deer, fallow deer, and Sika deer, the prevalence and intensity of infestation were directly proportional to the size of the host. Our findings are consistent with this conclusion since roe deer are smaller than red deer and fallow deer. Visual stimuli are considered important during the host location behavior of hippoboscids [57], and these parasites probably tend to attack larger species that are more easily detectable because of their size. In addition, several hematophagous ectoparasites use chemical cues (CO_2_ or odors) during the host-finding process [58]. Lourenço and Palmeirim [59] found that two Nycteribiidae species (Hippoboscoidea superfamily) mainly used carbon dioxide for long-distance host locations. This cue is emitted by all vertebrates, and animals larger in size tend to release it in larger amounts [60]. The antennae of *L. cervi* and *L. fortisetosa* are equipped with a well-developed sensory pattern on the external surface of the pedicel, suggesting that these different types of sensilla are likely able to perceive the chemical cues emitted by the hosts, supporting the hypothesis that more than visual signals alone are responsible for identifying host locations [6,61]. We can speculate that the large amount of CO_2_ released by the hosts may contribute to explaining how the roe deer are attacked less than the other two species. Haarløv [10] suggested that red deer species occur in habitats that are more suitable for the development of pupae that fall on substrates that are more suitable for their survival. In our case, red deer and fallow deer coexist in the study area and share the same territories, making it difficult to hypothesize that the local habitat strictly affects host choice. Most likely, instead, preference for red deer could be due to the physical features of this host species, such as the structure of its coat. In fact, the host fur represents the environment in which hippoboscids live, so it should have the conditions they need to survive. Red deer have long and robust guard hairs with a dense layer of underhairs at the base; however, roe deer and fallow deer have shorter hairs forming a softer but thicker covering that may obstructed parasites from reaching the skin, making trophic activity more difficult [10,61].

In this paper, the objective of verifying the possible differences between host age classes and sex was determined only for red deer since this species was the favorite host for both *L. cervi* and *L. fortisetosa*.

Kadulski [52] observed that the intensity of infestation increases with the size of a host. Our results show significant differences in the choice of host age classes by both deer keds. Other authors highlighted that fawns are attacked less than subadults or adults [38,56]. This preference could be due to fawn behavior since they follow dams during their first year of life. Given that fawns are together with the adult females, the flies are more likely to choose the larger subject since it is more visible and releases a greater amount of CO_2_. Regardless, body size alone cannot explain the host choice displayed by these hippoboscids; in fact, other aspects, such as the behavior and ecology of the host species, interact to affect this selection. Madslien et al. [38] hypothesized that *L. cervi* prefers parasitizing subadults over adult moose since the former moves more, increasing the chance of encountering a deer ked. Another explanation for this preference could be the resistance that some host species seem to develop toward hippoboscid attacks, as suggested for reindeer [62] and moose [38]. However, deer do not show similar resistance in the case of severe infestations [56].

According to our results, *L. cervi* is significantly more abundant in males. Additionally, in this case, the larger size and the more intense motility of host males may explain this preference, but it is also possible to hypothesize that odor secretions emitted by red deer can affect host choice as well. Deer have specialized regions with glands, whose activity may change between sexes, producing secretions to mark their territory. As demonstrated for white-tailed deer (*Odocoileus virginianus*), this glandular activity is higher in males, especially in dominant subjects [63]. Additionally, in *C. elaphus**,* there are quantitative and qualitative differences between males and females in terms of their released compounds [64]. As reported by Johnson and Leask [65] for *C. capreolus*, glandular activity and active testosterone metabolism can increase just prior to and during the mating season. In Italy, the breeding season of red deer occurs from late summer to early fall, overlapping with the host seeking period of *L. cervi*, possibly affecting the preference of the parasite for males. Further studies are needed to confirm the influence of sex differences in the odor secretions on *L. cervi* host selection.

Although *L. cervi* and *L. fortisetosa* are restricted to a limited group of species, they are able to adapt to new hosts and do not appear to strictly follow a parasitization scheme. In fact, we found both hippoboscids on all three examined deer species and on all host age classes and sexes. Apparently, these flies cannot be too selective in terms of sex and age classes since they are obligate ectoparasites and need to find a host shortly after emergence. The host species, however, seems to be an important prerequisite for *Lipoptena* spp.; in fact, red deer are favored by both flies.

Host density is one of the most important factors that needs to be considered when studying the distribution of hippoboscid ectoparasites, even if it does not explain all of the variation in the expansion of these flies. For instance, Meier et al. [35], suggested that a local increase in host density may allow for the rapid establishment of allochthonous ectoparasites. Additionally, *L. cervi* occurred in Finland in 1960 when the moose density experienced a large population growth [66]. The study of the relationship between parasites and hosts is fundamental, especially when it concerns allochthonous species which are able to adapt to new hosts, competing with native species/fauna. Hosts can represent the easiest transfer option for ectoparasites so that they can be disseminated in new territories during host movements and introductions. Just as the expansion of *L. cervi* in the northeastern United States is considered to be due to the anthropogenic introduction of European deer [9], it is likely that *L. fortisetosa* spread to Europe due to the relocation of its original host, *Cervus nippon*. However, the possible hybridization between sika and red deer, or the translocation of *C. elaphus*-related subspecies to Europe cannot be ignored. Currently, *C. nippon* is recorded in 20 European countries, while *L. fortisetosa* is present in 13 European countries [67]. In Italy, a great increase in cervid abundance has been recorded in recent years, and the presence of *C. nippon* has been recently documented [3,5]. This situation confirms the risk related to the increase in the abundance of native ectoparasites, together with the spread of alien parasitic species further favored by global warming.

## 5. Conclusions

The results of the present study show the great expansion of the allochthonous parasite *L. fortisetosa*, recently detected in Italy. This fly, originally restricted to the main host *C. nippon*, has a strong adaptability to other host species, such as red deer, fallow deer, and roe deer. Moreover, it seems to strongly compete with the autochthonous hippoboscid *L. cervi*, being more numerous in the study area. The favored host of both flies was red deer, even if all three examined host species harbored parasites. Different preferences for sex and age classes of the hosts were observed in the two hippoboscids. Although some explanations were hypothesized for these outcomes, at present, it is difficult to provide a specific explanation, since each choice occurred due to the interactions of many factors. Thus, further investigations are ongoing. Another aspect worthy of attention is related to the possible health risk implicated in the expansion of allochthonous species as potential vectors of harmful pathogens. Therefore, hippoboscid populations should be continuously monitored to promptly identify possible substantial expansion or adaptation to other host species, which can lead to further spread with negative consequences from both ecological and health perspectives. Regular monitoring of deer keds should also be carried out to improve the knowledge of these parasites and establish specific management strategies to limit hippoboscid expansion.

## Figures and Tables

**Figure 1 animals-11-02794-f001:**
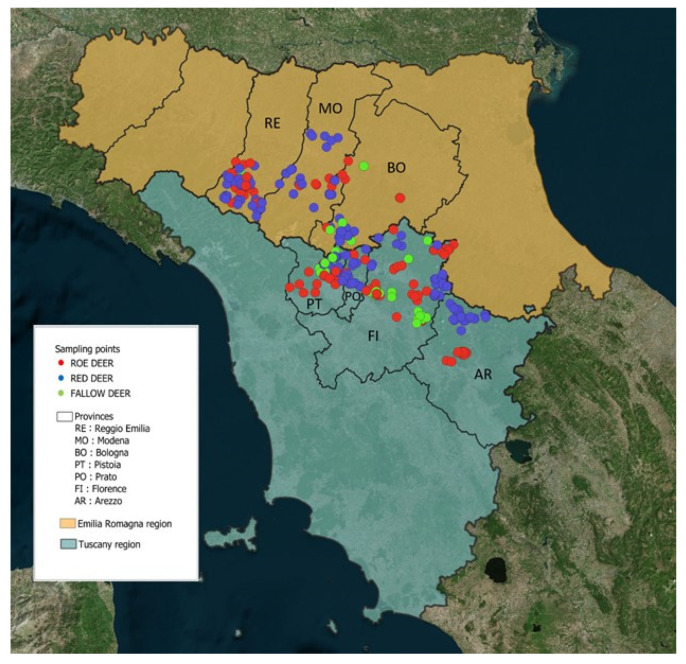
Map of host sampling sites in the Tuscan-Emilian Apennines (central Italy). (Map created using the Free and Open Source QGIS).

**Figure 2 animals-11-02794-f002:**
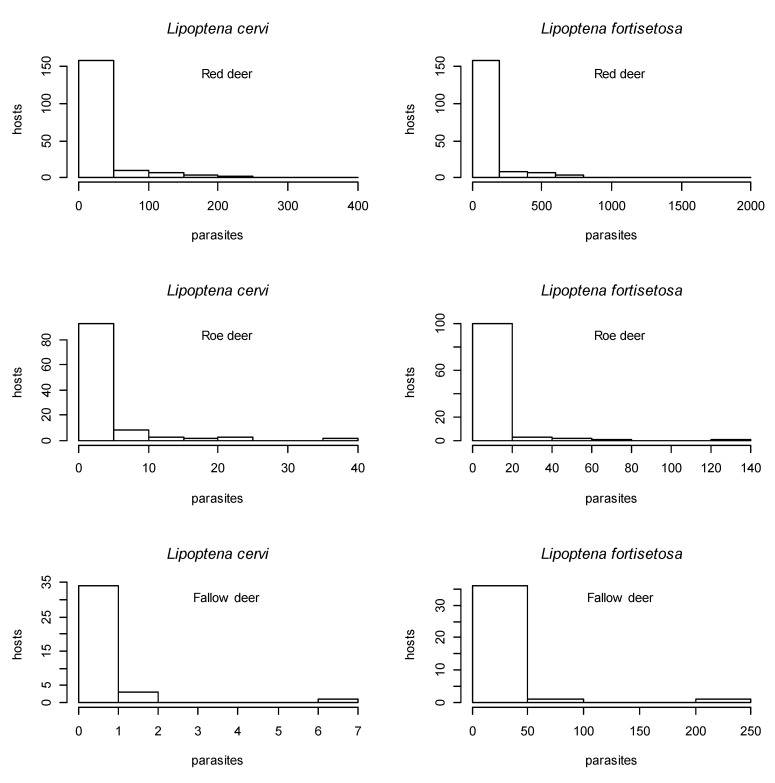
Histograms of parasite distribution: both parasite species are aggregated on all three host species.

**Figure 3 animals-11-02794-f003:**
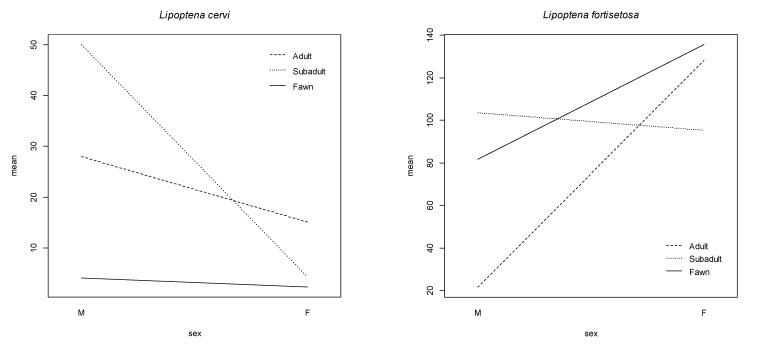
Interactions between sex and age in determining the mean abundance of *Lipoptena cervi* and *Lipoptena fortisetosa* on red deer.

**Figure 4 animals-11-02794-f004:**
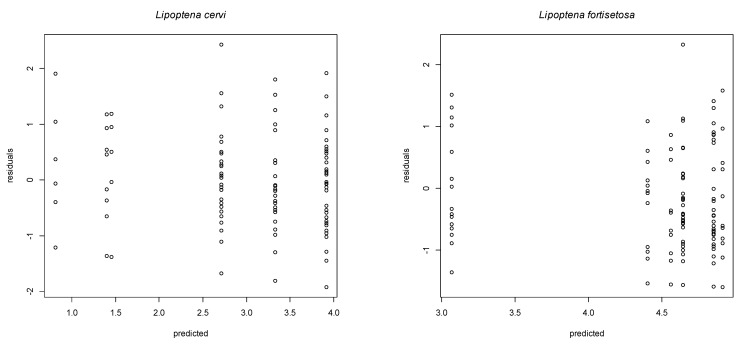
Plot of residuals vs. predicted values of the multivariable model with *Lipoptena cervi* as the dependent variable and red deer age and sex as covariates.

**Table 1 animals-11-02794-t001:** Epidemiological descriptors of *Lipoptena* spp. infestation on different hosts.

Host Species	Total Number of Deer Keds Collected on Infested Hosts	Density (n. Parasites/cm²)	Prevalence of Infestation % (n. Infested Hosts/n. Tested Hosts)	Prevalence of Mixed Infestation % (n. Hosts Infested by Both Parasites/n. Tested Hosts)	Mean Abundance (n. Parasites/n. Tested Hosts)	Mean Intensity of Infestation (±sd) (n. Parasites/n. Infested Hosts)	Intensity of Infestation (*I*min-*I*max)
Red deer (*n* = 181)	21,548	0.149	93.9	52.5	119.05	126.75 (±235.67)	0–1,844
Fallow deer (*n* = 38)	645	0.021	94.7	21.0	16.97	17.92 (±37.37)	0–214
Roe deer (*n* = 107)	881	0.010	75.7	22.4	8.23	10.88 (±16.04)	0–123

**Table 2 animals-11-02794-t002:** Epidemiological descriptors of the ectoparasites *Lipoptena fortisetosa* and *Lipoptena cervi* on their three main hosts in central Italy.

Deer Keds and Host Species	Total Number of Deer Keds Collected on Infested Hosts and Sex Ratio	Density (n. Parasites/cm²)	Prevalence of Infestation % (n. Infested Hosts/n. Tested Hosts)	Mean Abundance (n. Parasites/n. Tested Hosts)	Mean Intensity of Infestation (±sd) (n. Parasites/n. Infested Hosts)	Intensity of Infestation (*I*min-*I*max)	Aggregation Index (Variance/Mean Abundance)
*Lipoptena fortisetosa*							
Red deer (*n* = 181)	17,194 (6999/10,195)	0.118	71.3	94.99	133.28 (±243.88)	0–1,844	500 (>1)
Fallow deer (*n* = 38)	628 (331/297)	0.021	94.7	16.53	17.44 (±37.96)	0–214	83 (>1)
Roe deer (*n* = 107)	619 (345/274)	0.007	53.3	5.79	10.86 (±19.80)	0–123	50 (>1)
*Lipoptena cervi*							
Red deer (*n* = 181)	4354 (1610/2744)	0.030	75.1	24.06	32.01 (±57.39)	0–398	111 (>1)
Fallow deer (*n* = 38)	17 (9/8)	0.001	21.0	0.45	2.12 (±2.03)	0–7	3.5 (>1)
Roe deer (*n* = 107)	262 (109/153)	0.003	44.9	2.45	5.46 (±7.3)	0–37	12.5 (>1)

**Table 3 animals-11-02794-t003:** Epidemiological descriptors, stratified by host sex and age class, of the ectoparasites *Lipoptena fortisetosa* and *Lipoptena cervi* on their three main hosts in central Italy.

Deer Ked Species	Host Species and Sex	Age Class and Number of Sampled Hosts	Total Number of Deer Keds Collected on Infested Hosts	Density (n. Parasites/cm²)	Prevalence of Infestation % (n. Infested Hosts/n. Tested Hosts)	Mean Abundance (n. Parasites/n. Tested Hosts)	Mean Intensity of Infestation (± sd) (n. Parasites/n. Infested Hosts)	Intensity of Infestation (*I*min-*I*max)	Aggregation Index (Variance /Mean Abundance)
** *Lipoptenafortisetosa* **									
Red deer									
	female	*adult (n = 49)*	6274	0.16	71.4	128.04	179.26 (±293.29)	0–1,048	500 (>1)
		*subadult (n = 11)*	1050	0.12	81.8	95.45	116.67 (±158.88)	0–419	250 (>1)
		*fawn (n = 20)*	2714	0.17	60.0	135.70	226.17 (±397.09)	0–1,311	1000 (>1)
	male	*adult (n = 36)*	773	0.03	63.9	21.47	33.61 (±58.02)	0–195	100 (>1)
		*subadult (n = 49)*	5078	0.13	73.5	103.63	141.06 (±330.45)	0–1844	2.8 (>1)
		*fawn (n = 16)*	1305	0.1	87.5	81.56	93.21 (±133.68)	0–474	200 (>1)
Fallow deer									
	female	*adult (n = 8)*	73	0.01	100.0	9.13	9.13 (±15.06)	1–46	25 (>1)
		*subadult (n = 3)*	22	0.01	100.0	7.33	7.33 (±4.51)	3–12	2.8 (>1)
		*fawn (n = 0)*							
	male	*adult (n = 10)*	288	0.04	100.0	28.80	28.8 (±65.36)	1–214	142.9 (>1)
		*subadult (n = 12)*	203	0.02	91.7	16.92	18.45 (±27.75)	0–95	43.4 (>1)
		*fawn (n = 5)*	42	0.01	80.0	8.40	10.50 (±9.95)	0–23	11.1 (>1)
Roe deer									
	female	*adult (n = 25)*	67	0.003	36.0	2.68	7.44 (±13.46)	0-43	27.8 (>1)
		*subadult (n = 10)*	62	0.01	50.0	6.20	12.4 (±14.88)	0–37	22.7 (>1)
		*fawn (n = 9)*	3	0.0004	22.2	0.33	1.5 (±0.71)	0–2	1.5 (>1)
	male	*adult (n = 37)*	360	0.01	64.9	9.73	15 (±27.84)	0–123	50 (>1)
		*subadult (n = 14)*	80	0.01	78.6	5.71	7.23 (±7.78)	0–27	10 (>1)
		*fawn (n = 12)*	47	0.0005	50.0	3.92	7.83 (±7.68)	0–21	10 (>1)
** *Lipoptena cervi* **									
Red deer									
	female	*adult (n =* 49)	738	0.02	83.7	15.06	18 (±31.02)	0–171	50 (>1)
		*subadult (n =* 11)	47	0.01	36.4	4.27	11.75 (±6.60)	0–19	11.1 (>1)
		*fawn (n =* 20)	45	0.003	45.0	2.25	5 (±5.77)	0–19	10 (>1)
	male	*adult (n =* 36)	1006	0.03	77.8	27.94	35.93 (±54.88)	0–204	83.3 (>1)
		*subadult (n =* 49)	2453	0.06	91.8	50.06	54.51 (±80.48)	0–398	125 (>1)
		*fawn (n =* 16)	65	0.01	56.3	4.06	7.22 (±6.04)	0–18	7.7 (>1)
Fallow deer									
	female	*adult (n =* 8)	7	0.001	12.5	0.88	7	0–7	7.1 (>1)
		*subadult (n =* 3)	0						
		*fawn (n =* 0)							
	male	*adult (n =* 10)	2	0.0003	20.0	0.2	1	0–1	0.91 (<1)
		*subadult (n =* 12)	8	0.001	41.7	0.67	1.6 (±0.55)	0–2	1.2 (>1)
		*fawn (n =* 5)	0						
Roe deer									
	female	*adult (n =* 25)	147	0.01	52.0	5.88	11.31 (±11.4)	0–37	16.7 (>1)
		*subadult (n = 10)*	15	0.002	30.0	1.5	5 (±4.58)	0–10	7.1 (>1)
		*fawn (n =* 9)	8	0.001	55.6	0.89	1.6 (±1.34)	0–4	1.8 (>1)
	male	*adult (n =* 37)	46	0.002	43.2	1.24	2.87 (±3.03)	0–10	4.8 (>1)
		*subadult (n =* 14)	22	0.002	35.7	1.57	4.4 (±3.29)	0–8	5.3 (>1)
		*fawn (n =* 12)	24	0.003	50.0	2	4 (±3.69)	0–11	5.3 (>1)

**Table 4 animals-11-02794-t004:** Results of the multivariable negative binomial model with *Lipoptena cervi* as the dependent variable and red deer age and sex as covariates.

*Lipoptena cervi*	Coefficient	Std. Error	z	*p* Value
Intercept	3.913	0.233	16.801	0.000
Sex				
Male	Reference			
Female	−2.461	0.563	−4.382	0.000
Age				
Subadult	Reference			
Adult	−0.583	0.359	−1.626	0.104
Fawn	−2.511	0.484	−5.186	0.000
Interactions				
Adult-Male	Reference			
Adult-Female	1.843	0.667	2.763	0.006
Fawn-Female	1.870	0.806	2.320	0.020

**Table 5 animals-11-02794-t005:** Results of the multivariable negative binomial model with *Lipoptena fortisetosa* as the dependent variable and red deer age and sex as covariates.

*Lipoptena fortisetosa*	Coefficient	Std. Error	z	*p* Value
Intercept	4.641	0.324	14.324	0.000
Sex				
Male	Reference			
Female	−0.082	0.757	−0.109	0.914
Age				
Subadult	Reference			
Adult	−1.574	0.499	−3.155	0.002
Fawn	−0.240	0.653	−0.367	0.714
Interactions				
Adult-Male	Reference			
Adult-Female	1.868	0.906	2.061	0.039
Fawn-Female	0.591	1.073	0.551	0.582

## Data Availability

No data were deposited in an official repository.

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
