# Peer review of "Distribution of Deer Keds (Diptera: Hippoboscidae) in Free-Living Cervids of the Tuscan-Emilian Apennines, Central Italy, and Establishment of the Allochthonous Ectoparasite Lipoptena fortisetosa"

_animals, 2021, doi:10.3390/ani11102794_

Round 1

Reviewer 1 Report

I revised the manuscript entitled “Distribution of deer keds (Diptera: Hippoboscidae) in free-living cervids of the Tuscan-Emilian Apennines, central Italy, and establishment of the allochthonous ectoparasite Lipoptena fortisetosa " by Annalisa Andreani and collaborators.

General comment:

The manuscript is interesting, and its topic is of interest to those who are working in the field. The aim of the study was to investigate the parasitism level of both L. fortisetosa and L. cervi on their main hosts in Italy (i.e., red deer, fallow deer, and roe deer); further aims were to highlight possible preference for species, sex, or age class among the hosts.

The manuscript reports some novelties and is well structured; the introduction and material and methods section are clear, complete, and well-structured, methods are adequate and presented in a clear way; results are consistent and clearly presented. The discussion section is solid and well structured.

Pending minor revision, in my opinion, the Ms is suitable for your Journal

Below you find a few specific comments, that in my opinion, could ameliorate the Ms.

Specific comment:

Lines 53-76: In my opinion, this part is too long and too didactic and non-strictly functional to the Ms, I suggest shortening this period.

Line 90: Please into “to a very high level” into “heavily”.

Lines 96-97: This sentence is redundant with the one reported in lines 90-92 please summarize it.

Lines 190: I suggest changing the paragraph title into “parasitological index” and the same comment could be applied to line 194.

Lines 214: Please specify if all the animals were positive to the presence Lipoptena spp.; if no, please indicate the number of positive animals and the relative percentage. The Authors observed mixed infections?

Lines 216-218: This sentence is not very clear please rewrite clearly.

Line 219: I suggest indicating that the number of males and females for each species and the sex ratio is reported in Table 1

Line 222-223: Are these differences observed statistically significant? Please report the P-value

Line 231: Please specify what this P value refers to.

Lines 232-233: The difference is statistically significant?

Tables and figures: Please uniform in all the tables and in all the figures the name of the hosts (always use or the common name or the scientific name) and report Lipoptena spp. in extenso and in Italics.

Results: Any results about the presence/abundance in the different provinces of the study site are reported. Please add some information

Line 289: Please change into “C. capreolus” and “D. dama”

Line 296-297: Considering that in the results section any information about the distribution of the two different species is reported how the authors discuss the simultaneous presence of the two parasites? I guess that in all the province both Lipotena species were observed, but before to discuss this point is important to provide the results. Moreover, if exist a difference in the abundance of the two species in the different provinces in which the animals were hunted will be interesting to put in relation also the environment with the parasite presence.

Line 333: Please better define “cue”, maybe its better to modify it into “CO2”?

Reviewer 2 Report

This manuscript reports a study on the occurrence (frequency and burden) of two species of Lipoptena keds in deer of three species in central Italy with L. cervi representing the well-known common deer ked in Europe while L. fortisetosa represents a species which is thought to originate from the eastern Palearctic and has been detected in central Europe only in the 20th century. The subject of the manuscript is relevant as it provides further insight in the spread of L. fortisetosa and on the relevance of deer keds in general. The manuscript is one more contribution of this working group and provides valuable information towards the study of deer keds in Italy (and Europe).

However, several revisions are to be made before the manuscript should be accepted for publication. In general, the manuscript would benefit if reviewed by a native speaker and Introduction and Discussion chapters would be shortened specifically for the details of potential public/human health implications as these is not only not the topic of this study but are further in most aspects not sufficiently supported.

Specific aspects requiring attention:

Simple Summary talks about incidence (which is likely prevalence meant with reference to the use of Margoli’s definitions) and considers L. cervi as Palearctic and L. fortisetosa as allochthonous. Both species are Palearctic but L. fortisetosa is allochthonous in contrast to L. cervi – clarification/correct wording is needed.

It is further said that L. fortisetosa is native in Japan; however, it is relevant to state that this is also the case for mainland Eastern Asia (China, Russia).

Abstract

It is not correct to state that keds were collected from 326 deer: samples of 326 deer were examined and keds were collected from 253 deer actually!

Knowledge on keds is not to be improved but rather to be complemented, supplemented or expanded.

Introduction

See comments made for Simple Summary.

Lines 105 etc. To the best of the reviewer’s knowledge, the vectorial capabilities of keds are not that well understood. The detection of zoonotic pathogens in a hematophagous insect does not constitute prove of its role in the transmission of the pathogen. It is, for sure, a prerequisite but is no prove at all. Thus, the authors need to say correctly that the presence of vector-associated pathogens has been reported which suggest that keds may be vectors unless they are able to clearly identify and refer to studies/publications which proved the vector role. In that context, reference to One Health etc. should preferably not be made as long there is no evidence.

Material and Methods

In order to add value as to the epidemiology of the keds infestation, the authors should provide information as to the months in a year which represent the period(s) of collection of the keds.

It is stated that the skin samples were examined for both keds and ticks; however, only ked data are given. Authors may add information as to the tick infestation recorded.

Results are presented by ked species and host species (see Table 1). However, in order to allow for a complete understanding of the infestation by keds, a presentation of both ked species combined by host needs to be provided. Otherwise no judgement of the total ked infestations burden by species is possible.

With reference to Barbour & Pugliese, the authors claim the calculation of the variance-mean-to-mean (abundance) ratio as measure for the aggregation; however, they calculated the mean (abundance)-to-variance ratio. While the interpretation of the outcome is not in question, this “discrepancy” should be reconciled and the referred to publication should be followed.

Results

As indicated for the Abstract, wording should be clarified in that the number of ked-positive animals is identified from which the keds were actually collected and there is a strong need to add information on how many hosts per species carried mono and mixed infestation and, in order to allow for the assessment of “overall ked infestation”, Table 1 data for combined L. cervi and L. fortisetosa needs to be added (including information on maximum/minimum combined ked counts).

Importantly, results/ked counts are presented as number “collected from a host specimen” or “picked off a single deer” – however, actually 800 cm² skin was examined leading to the question what count does represent the numbers given (total count per host “extrapolated” from 800 cm² count or just the 800 cm² count which seems unlikely) – clarification is needed, already in the Materials and Methods section!

For Table 1, there is the “Density” as number of keds per cm” is given, for red deer L. fortisetosa 42.99 keds per cm² - this is highly unlikely. Checking of numbers, clarification and explanation needs to be provided! (this applies to Tab. 2 too).

Comments on the aggregation index were provided already and should be considered accordingly.

Discussion

Reference on the co-occurrence of L. fortisetosa and H. longipennis on dogs appears not to be appropriate in the context of the two Lipoptena species in deer as the dog is unlikely to be a breeding host for L. fortisetosa.

Minimum “overall” fly infestation could not be 21% only but should be actually 75.1% (red deer).

Clarification on the density figures is needed.

“Per” single bull or in a single bull?

Intensity of infestation appears to be correlated with host “size” rather than host “growth”.

  1. fortisetosa is for sure originally an element of the Eastern Palearctic; however, whether it unlikely derived exclusively from sika deer of Japan only but likely also from sika deer inhabiting the mainland of East Asia (Far East of Russia, China) and potentially other Cervus elaphus-related subspecies of deer which were translocated into the European part of the former Soviet Union and are likely the source of spreading of L. fortisetosa at least for the eastern part of Europe.
